# Alpha interbrain synchrony during mediated interpersonal touch

**Wanjoo Park**[ID]**, Muhammad Hassan Jamil, Mohamad Eid**[ID]*

Engineering Division, New York University Abu Dhabi, Saadiyat Island, Abu Dhabi, United Arab Emirates

* mohamad.eid@nyu.edu

**Data Availability Statement:** The data that support the findings of this study are openly available at the following URL: https://osf.io/dpq8f.

**Funding:** YES. M.E. received the funding. This work is supported by the NYUAD Center for Artificial Intelligence and Robotics (CAIR), funded

## Abstract

Interpersonal touch plays a crucial role in human communication, development, and wellness. Mediated interpersonal touch (MIT), a technology to distance or virtually simulated interpersonal touch, has received significant attention to counteract the negative consequences of touch deprivation. Studies investigating the effectiveness of MIT have primarily focused on self-reporting or behavioral correlates. It is largely unknown how MIT affects neural processes such as interbrain functional connectivity during human interactions. Given how users exchange haptic information simultaneously during interpersonal touch, interbrain functional connectivity provides a more ecologically valid way of studying the neural correlates associated with MIT. In this study, a palm squeeze task is designed to examine interbrain synchrony associated with MIT using EEG-based hyperscanning methodology. The phase locking value (PLV) index is used to measure interbrain synchrony. Results demonstrate that MIT elicits a significant increase in alpha interbrain synchronization between participants' brains. Especially, there was a significant difference in the alpha PLV indices between no MIT and MIT conditions in the early stage (130–470 ms) of the interaction period (t-test, $p < 0.05$). Given the role that alpha interbrain synchrony plays during social interaction, a significant increase in PLV index during MIT interaction seems to indicate an effect of social coordination. The findings and limitations of this study are further discussed, and perspectives on future research are provided.

## Introduction

The human sense of touch is the most intimate form of interpersonal communication [1]. Interpersonal touch has been shown to play a crucial role in social interaction [2], emotional communication (such as kissing, hugging, hand-holding, and cuddling) [3], childhood/adulthood development and well-being [1], relationship formation and maintenance [4], and stress [5] and pain [6] management. For instance, interpersonal touch, such as a strong handshake or a nudge for attention, can convey vitality and immediacy more powerfully than language [1]. Furthermore, touch is commonly utilized to enhance the meaning of other forms of verbal and non-verbal communication (e.g. touch amplifies the intensity of emotional response from our face and voice [7]) [8]. Interpersonal touch is also known to provide a powerful means of compliance gaining (the Midas touch effect [9], for example a customer gives bigger tip when

by Tamkeen under the NYUAD Research Institute
Award, CG010. URL for funding agency: https://
www.tamkeenuae.com/.

**Competing interests:** The authors declare that the
research was conducted in the absence of any
commercial or financial relationships that could be
construed as a potential conflict of interest.

slightly touched by a server). On the other hand, touch deprivation often carries negative physiological, cognitive, social, communication, and emotional connotations, such as at early stages of human development [10] or during a pandemic [11].

Although touch is a critical component of interpersonal communication, its implementation in computer-mediated communication is yet to be fully realized. Recent research into interpersonal touch, as well as advances in haptic technologies, have inspired the development of technology-mediated interpersonal touch (MIT) [12], with the aim to simulate interpersonal touch with a virtual agent or a remote user. Research in MIT resulted in a wide range of technologies with different design philosophies and target applications, ranging from a wearable sleeve [13], wristband [14], glove [15] to a hugging vest [16, 17] or even a full bodysuit [18]. These systems offer interesting interaction possibilities and demonstrate the technical feasibility of MIT. Compared to direct interpersonal touch, MIT can induce feelings of intimacy and sympathy [19], connectedness toward another person [20, 21], prosocial behavior [22], and emotional communication [3].

What is still obscure is how to quantitatively evaluate the effectiveness of MIT technologies [23]. Existing techniques are mostly based on either self-reporting (using questionnaires or think-aloud protocols) or behavior evaluation such as eye contact, facial expressions, gestures, and physiological measures during the interaction. Neurophysiological methods such as functional magnetic resonance imaging (fMRI) and electroencephalography (EEG) provide a quantitative measure of the subject's mental state without disturbing the interaction [24]. Compared to fMRI, EEG is a lower-cost apparatus that is capable of recording the cortical brain's electrical activity in a more natural environment, which is crucial for interpersonal interactions. EEG also has high temporal resolution, and thus it allows for real-time analysis of the neural mechanisms of interpersonal touch. A number of studies examined neural responses to direct [25, 26] or mediated [27] interpersonal touch. A study indicates that hand-holding between romantic partners during pain increases inter-brain synchrony and eventually induces analgesia [28]. A recent study demonstrated that hand-holding with a partner (compared to touching by a stranger or being alone) augments theta and beta power as an indication of lower emotional arousal [29].

Given that interpersonal touch represents a complex system where each subject influences and is influenced by the other, the coordinated neural activity can be explored using interbrain dynamics techniques [30]. Hyperscanning is a neuroscience technique to obtain simultaneous neural recordings from more than one person in order to study interbrain synchrony (using fMRI such as [31] or EEG such as [32]). EEG-based hyperscanning is becoming increasingly popular since it allows researchers to explore interbrain synchrony in more natural settings such as by allowing mobility and high temporal recording resolution. For instance, hyperscanning studies in subjects simultaneously engaged in a card game [33] or in tasks that are related to game theory [34–36], provide evidence that mathematical indices that are derived from multiple-brain modeling can discriminate cooperative and individualistic behaviors. The phase locking value (PLV) is a measure of the phase synchrony between two time-series, which has been previously applied to examine EEG interbrain functional connectivity between different regions in the brain or even between two brains [37]. If the phases of the two signals are strongly coupled then the PLV value will approach the unit.

The inter-brain mechanisms that underlie MIT are largely unknown. In this study, we apply a hyperscanning approach with mediated hand squeeze interaction of dyads to examine the association between brain-to-brain coupling and MIT. Our findings indicate that MIT increases interbrain synchrony as indicated by a significant increase in the alpha band PLV index, which is typically correlated with social coordination. These findings make a unique contribution to our understanding of MIT.

## Materials and methods

### Participants

A total of sixty (60) healthy young adults participated in this study. These included 30 females and 30 males, with an age range of 18–24 years. Note that the total sample size is 30 dyads. Inclusion criteria were: adults aged 18 years and older and right-handed. Exclusion criteria included participants below the age of 18 or left-handed. All participants were healthy, with no prior physical or mental illness as confirmed through self-reporting. Participants in the same experimental session were recruited separately to make sure they were unfamiliar with each other. To ensure unfamiliarity, participants were separated by a curtain during EEG preparation so that their first encounter occurred during the experiment. The purpose was to reduce any potential bias since we anticipate a certain level of cooperation between the participants during the experiment. If two individuals were familiar with each other, it may influence the execution of the task, as one participant may unconsciously apply less or more pressure on the sleeve, knowing that the other person is their friend or acquaintance.

The study was conducted after obtaining approval on the experimental protocol by New York University Abu Dhabi Institutional Review Board (IRB: #HRPP–2020–11). The study was conducted in full compliance with the ethical standards outlined in the Declaration of Helsinki, following its guidelines and regulations. Before enrolling in the study, each participant signed an informed written consent form in accordance with the IRB ethics. All participants received monetary compensation at the end of the experimental session. No identifying information was collected from the participants.

### Experimental setup

A block diagram of the experimental setup is shown in Fig 1. The setup included a 128 active electrodes EEG system (an actiCHamp amplifier with the Brain Vision Recorder Version 1.21.0201, Brainproducts GmbH, Germany), a computer hosting the EEG recording software, a computer running the experimental application, two computer screens, two speakers, two keypads, and two cuffs (one for each participant), and a pressure control system. Each participant used a 64-electrode EEG cap where both caps were connected to the same EEG amplifier with common ground. The computer screens provided visual instructions to the participants. The speakers were used to render auditory cues during the experiment. The keypad was used to receive a response (yes/no) from participants about whether they felt the squeeze or not.

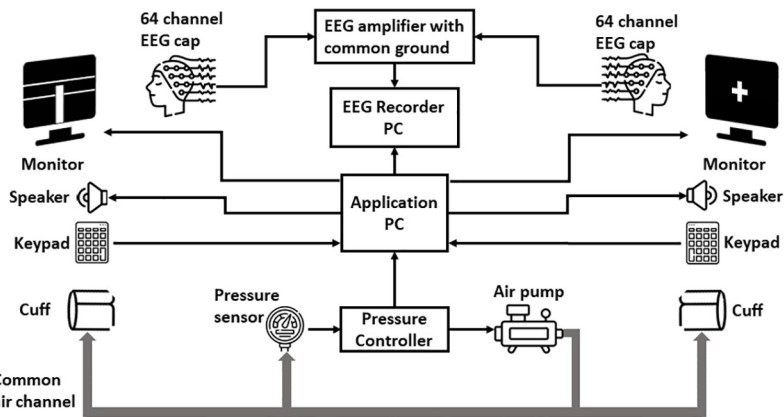

**Fig 1. A schematic diagram of the experimental setup.**

The squeezing system was implemented using a medium-size cuff (similar to a blood pressure apparatus cuff) attached to both participants' right hands. An air pump was used to inflate the cuffs up to 15 psi pressure level. Both cuffs were connected through the common air channel to let the air pressure transfer from the MIT encoder to the MIT decoder during the squeeze task thus giving a mild massage sensation on the MIT decoder palm. A feedback air pressure controller was established using a microcontroller unit (Arduino Uno) and a pressure sensor to maintain a uniform pressure level and to compensate for any air leaks during the experiment. The air pump was placed inside an acoustic container and away from the participants in order to minimize external noise. It is to be noted that the pressure controller was only activated between subsequent trials of the experiment to minimize interference with the recorded pressure measurements.

A software application was developed using the Unity 3D (Unity Technologies, United States) game engine to provide stimulation to both participants. The application was also responsible for recording and storing the pressure profile of the participants during the experimental task in a data file. The application controlled the two screens providing visual information to the encoder and the decoder. The encoder screen showed a vertical bar to indicate the squeeze pressure in real time. The encoder screen also displayed two horizontal lines representing the minimum and maximum squeeze pressure levels that the encoder was instructed to maintain the squeeze pressure in between. The minimum and maximum squeeze pressure levels were relaxed and used to make sure that the squeeze pressure was perceivable by the decoder while suppressing high variability in the squeeze behavior. The decoder screen displayed a fixation so the decoder focuses on perceiving the squeeze pressure via the cuff attached to the palm. The application was also used to send triggers to the EEG amplifier to mark the start and end of each phase of the trial.

## The experimental protocol

Participants were recruited via the Internet and by ads posted on the university campus. After signing the written informed consent form, participants were instructed about the experimental protocol and how to use the cuff interface. The experiment facilitator checks whether the participants fully understood and performed the task properly throughout the training session. Participants sat against each other at a distance of around four feet with a curtain separating them as shown in Fig 2. Once training was complete, a 64-electrode cap was placed on each

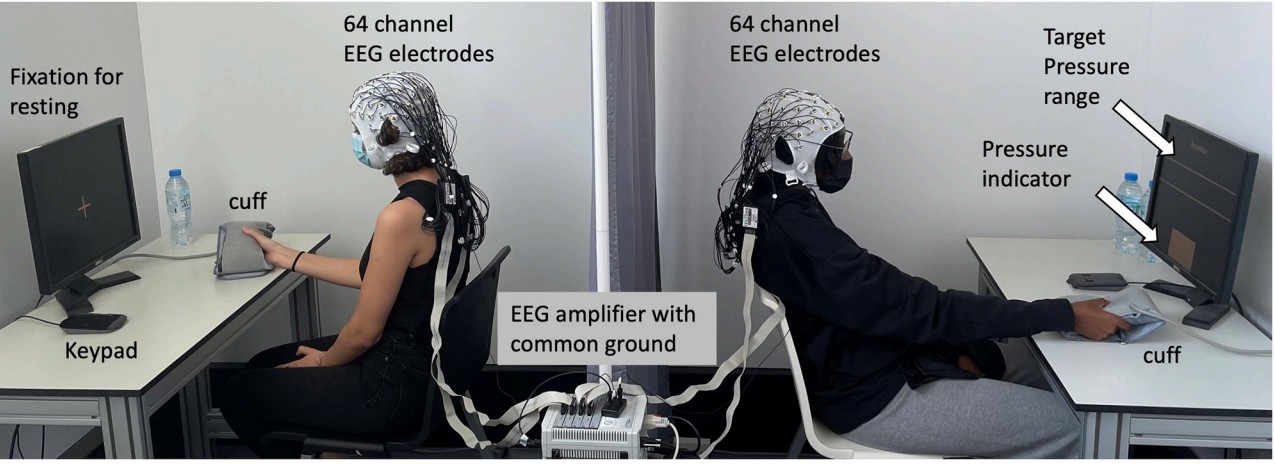

**Fig 2. A snapshot of the experimental setup.**

participant's scalp and the cuff was attached to the participant's right hand to restrict arm movements. The EEG electrodes were arranged according to the international 10–20 system. The ground and reference electrodes were located on Fpz and FCz. To ensure high-quality signal recordings, a connection impedance between the electrode and the scalp was maintained below 10 kΩ for all the electrodes.

The experimental protocol was divided into 7 runs separated by 2-minute rest periods. Each run was composed of a total of 18 trials (3 tasks × 6 repetitions). The three tasks were (1) participant 1 acts as the MIT encoder (transmits the squeeze) while participant 2 acts as the MIT decoder (feels the squeeze), (2) swapped roles where participant 2 is the MIT encoder while participant 1 becomes the MIT decoder, and (3) a neutral task where no squeeze (no MIT) is communicated between the two participants. Thus, there were 42 trials for each task. The repeated tasks were performed in a counter-balancing order through two 3 by 3 Latin squares [38].

As shown in Fig 3, a trial consists of rest, a task, and a question. While showing fixation to both participants, the rest period was randomized to 1000 ms or 2500 ms in order to minimize the prediction effects. The task started with an auditory cue of a 1000 Hz beep sound where the target pressure range (displayed as horizontal lines on the screen) and the pressure bar appear on the screen of the MIT encoder with instructions to squeeze the cuff within the desired pressure level range. At the end of the 2500 ms task period, a 500 Hz beep sound indicated that the task was over, and the screen of the MIT encoder displayed fixation again. And

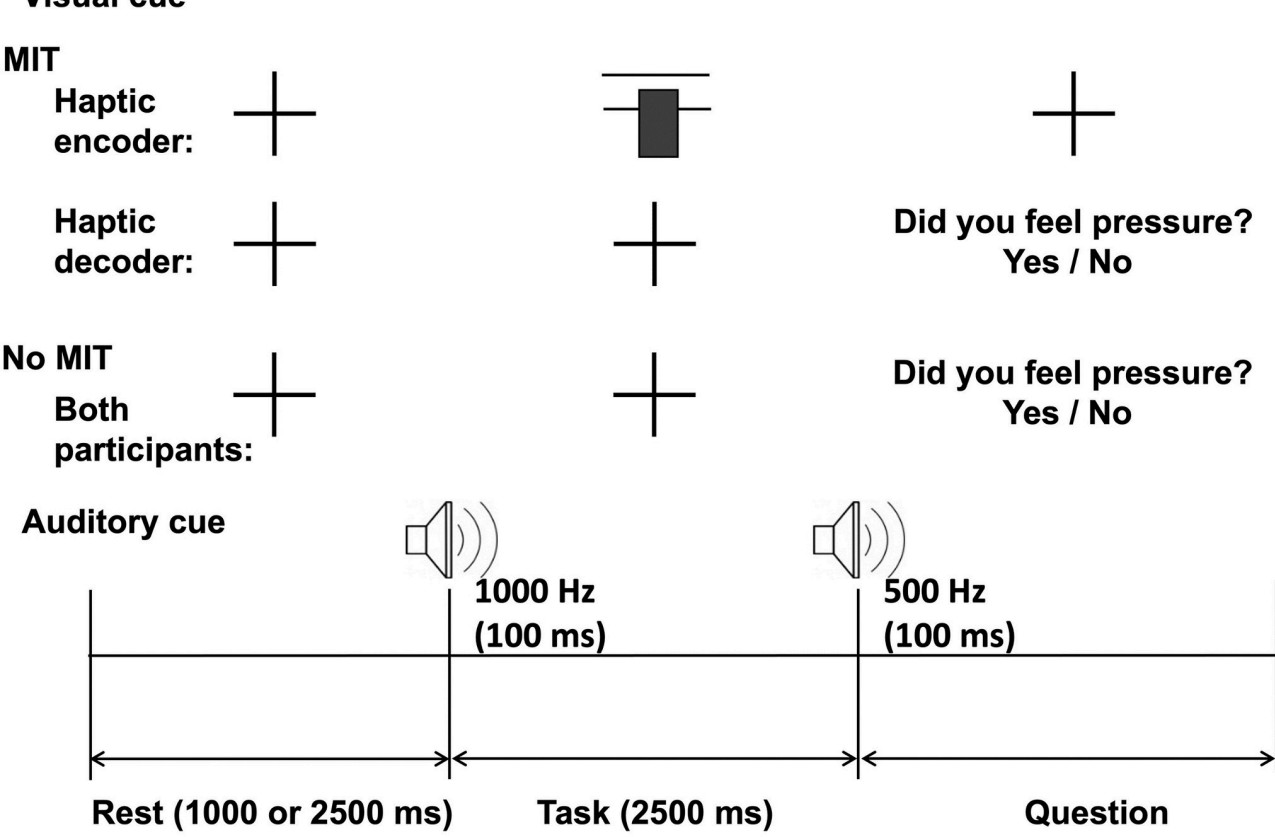

**Fig 3. Schematic representation of a single trial of the experimental task.**

then, the screen of the MIT decoder showed the question "Did you feel pressure?" and the MIT decoder answered 'yes' or 'no' using the keypad. The neutral task followed the same rest, task, and question periods and fixation appeared during the rest and task periods on the screens of both participants, and in the question period, the question "Did you feel pressure?" was shown to both participants, thus both of them had to respond before the next trial proceeds.

## Behavioral analysis

The behavioral data analysis aimed to (1) confirm that the squeeze message is properly communicated and is perceivable to the receiver, (2) analyze how the MIT communication affects the squeeze behavior, and (3) identify potential time of interest to conduct the neural analysis. The squeeze pressure data collected during the experiment was used to analyze the behavior of the participants during the task period. MATLAB was utilized to process the recorded pressure data [39]. Pressure data was recorded for each trial during the task time period of 2500 ms. The recorded data was adjusted for a baseline minimum pressure during task time as a preprocessing step. A low-pass Butterworth filter with a filter order of 12 and a half-power frequency of 0.04 was used to filter any high-frequency components from the pressure data.

To identify the onset time, the average pressure for the first 100 ms after the onset was used as a baseline. The onset was defined as the time at which the pressure level increases by more than a threshold from the baseline (the average pressure for the first 100 ms). The threshold was calculated empirically to be at 0.01 psi, which makes the pressure perceivable to the decoder. Peak pressure was marked by calculating the maximum applied pressure during the task period.

Three parameters related to the pressure profile were considered to analyze the squeeze behavior, namely the squeeze onset, the squeeze energy, and the squeeze speed. The squeeze onset was defined as the time at which the squeeze pressure becomes perceivable to the decoder. The squeeze energy was defined as the area under the squeeze pressure curve starting from the onset time until the end of the squeeze period (2500 ms). Finally, the squeeze speed was defined as the rate at which the squeeze pressure increases and was calculated by taking the difference in the squeeze pressure from the onset time until the peak divided by the time interval from the onset to the peak pressure.

## EEG analysis and functional connectivity

The EEG data was pre-processed using MATLAB release 2021a (MathWorks, United States) and the EEGLAB toolbox [40]. The outermost four EEG channels (FT9, FT10, TP9, and TP10) were excluded from the analysis. A zero-phase finite impulse response filter with a Hamming window was used for bandpass filtering (0.1–49.5 Hz). The artifact subspace reconstruction method (flat line criterion, 5; high pass filter, [0.25 0.75]; channel criterion, 0.8; line noise criterion, 4; burst criterion, 3; window criterion, 0.5) [41] was applied to remove eye movement and muscle artifacts. Then, EEG signals were re-referenced using the common average reference method [42]. The filtered EEG signal was epoched from -1,000 to 3,000 ms corresponding to the onset of the three tasks while 500 ms before the onset was used as the baseline. After pre-processing, the power spectral density (PSD) of each 1 Hz interval frequency bin at each channel was calculated via the short-time Fourier transform with a 500 ms Hamming window and a sliding step of 50 ms.

Topography analysis was performed for each frequency band, namely theta (4–7 Hz), alpha (8–12 Hz), beta (13–30 Hz), and gamma (31–49 Hz). In particular, changes in the PSD over time across the whole brain were analyzed in three conditions (no MIT, MIT encoder, and

MIT decoder). The no MIT communication was used as a control condition to investigate whether the PSD changes were due to the haptic interaction.

PLV index was used to investigate the EEG-based functional connectivity between the MIT encoder and decoder [37]. PLV indices of each frequency band were calculated according to the combinations of EEG channels between the MIT encoder and the MIT decoder. In order to determine the time of interest (TOI) during the task period (2500 ms), we investigated whether each time point in which the average PLV indices for all the EEG channel combinations between the MIT encoder and the MIT decoder are significantly different between no MIT and MIT (Wilcoxon signed-rank test and false discovery rate correction were considered). The normalized PLV indices were then calculated during the TOI for each combination of two participants' EEG channels. Statistical analysis also was utilized to see how differences in the PLV indices in TOI between the no MIT and the MIT conditions. We also investigated how differences in the PLV indices of two participants in TOI compare to the rest period.

## Results

### Behavioral evaluation

The grand average pressure profile as perceived by the MIT decoder is shown in Fig 4. The average onset timing was marked at 372 ms, at which the pressure becomes perceivable to the MIT decoder. The peak of the grand average pressure was observed by the MIT decoder at 1044 ms. A large standard deviation was observed towards the end of the squeeze task period as shown in Fig 4. This is related to the individual differences among MIT encoders when squeezing the cuff.

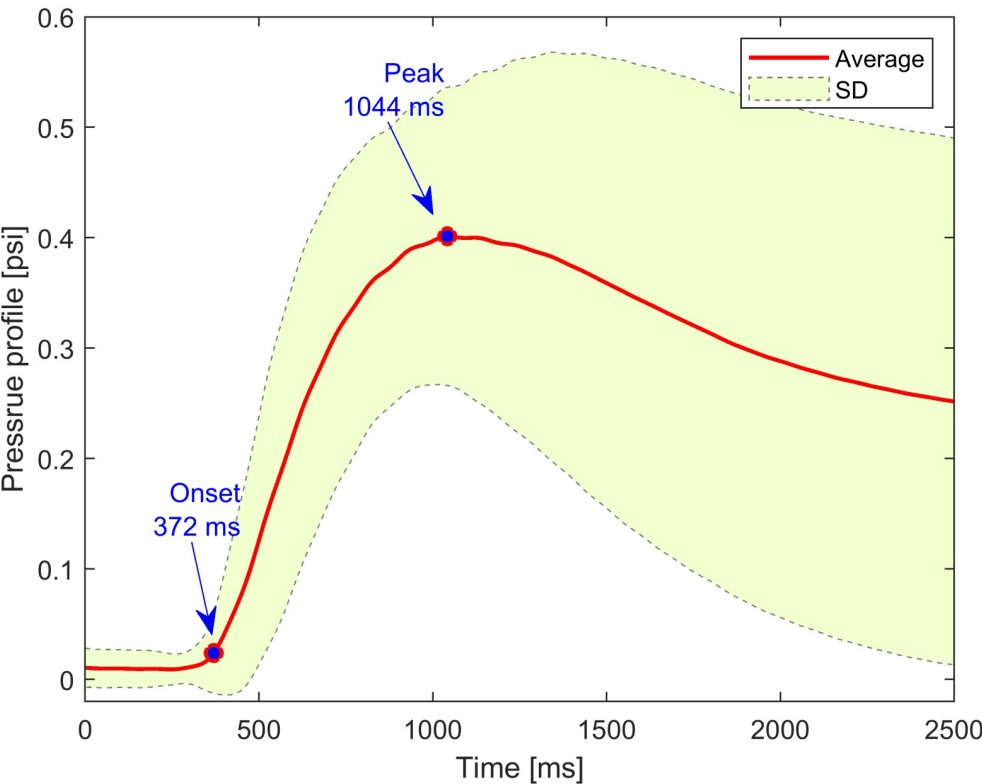

**Fig 4. Grand average pressure profile applied during the task period.**

In order to evaluate changes in the squeeze behavior over the experimental session, the differences between the two participants in the squeeze energy, squeeze onset, and squeeze speed, averaged over the first four and the last four trials are examined. The difference in the squeeze energy between the two communicating participants has decreased by around 6% by the end of the experimental session as compared to the beginning of the session. A similar change in the squeeze onset behavior is observed where the difference between the average squeeze onset of the two participants decreased by around 10% from the first four to the last four trials.

The MIT is well perceived by the communicating participants. When asked whether they perceived the squeeze or not, the MIT decoder confirmed that they felt the squeeze cue more than 95% of the times (Average, 98.28%; SD, 2.91%). Furthermore, during the no MIT task, both participants confirmed that no squeeze is felt more than 95% of the time (Average, 95.95%; SD, 6.74%)). This clearly demonstrates that the squeeze MIT was reliably communicated and perceived between the two participants.

## Topography analysis

Theta, alpha, beta, and gamma frequency bands were considered to investigate differences between the two conditions (no MIT and MIT). No significant differences are observed in any of the frequency bands except the alpha band. Fig 5 shows alpha PSD topographies during the task period calculated every 200 ms interval. First, in the no MIT task, no visible change in the alpha PSD is observed. However, within 400 ms, the alpha PSD of the frontal cortex slightly increases, presumably because the start of the task is recognized. For the MIT encoder, the prefrontal alpha PSD increased within 400 ms. It is presumed that prefrontal alpha oscillation increases due to recognizing the role of the MIT encoder and preparing for the motor task [43]. The alpha PSD topography of the MIT decoder at the same time (0–400 ms) is very similar to the no MIT interaction condition. The brain of the MIT encoder shows event-related desynchronization (ERD) after 400 ms. From 400 ms to 800 ms, there is stronger ERD in the contralateral hemisphere than in the ipsilateral hemisphere, followed by bilateral ERD activation, and finally, the ERD activation weakens over time. The squeeze onset (around 372 ms) implies that ERD was observed after 400 ms on PSD topography. Despite the fact that MIT decoder may feel the cuff inflate while the MIT encoder squeezes the cuff, there was no significant difference in the alpha PSD topography of the MIT decoder before 1000 ms. From the behavioral data, it takes about 1000 ms after the motor task cue for the air pressure of the cuff to reach the maximum (Fig 4), and it is shown that the alpha oscillation of the contralateral somatosensory cortex of the MIT decoder decreases starting from 1000 ms after the cue. In the interval from 1000 to 1400 ms, the decrease in the alpha oscillation of the contralateral somatosensory cortex is greater than that of the ipsilateral somatosensory cortex, and after 1400 ms,

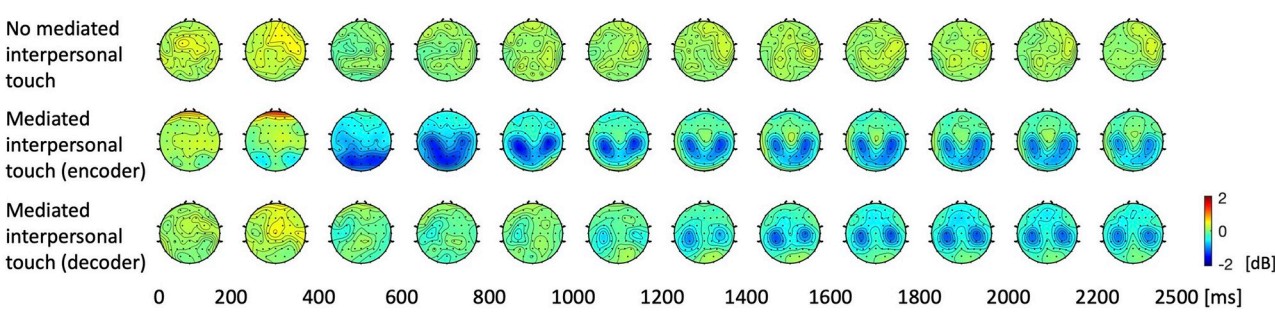

**Fig 5. Topographies of alpha power spectral density during the task period.**

the alpha oscillation in the bilateral somatosensory cortex is similarly decreased. This continues until the end of the task period.

### Interbrain functional connectivity

From the behavioral data, it is seen that the air pressure becomes perceivable after the squeeze onset (around 372 ms), and in the topography result, the change in the alpha oscillation of the MIT decoder decreases after the peak of the squeeze pressure (about 1044 ms from the onset). The functional connectivity between the MIT encoder and the MIT decoder is investigated using the alpha PLV index for the task period. Fig 6 shows how the alpha PLV index changes compared to the baseline (-100 to 0 ms prior to the cue), demonstrating the functional connectivity between the MIT encoder and the MIT decoder. The average of the total 3481 combinations of the two 59 EEG electrodes of the MIT encoder and the MIT decoder over all participants is shown in Fig 6. After the cue, the grand averages of normalized interbrain alpha PLV index increased in both of the no MIT and MIT conditions. However, the grand average in MIT condition increased more after around 100 ms. It peaks at around 270 ms and then gradually decreases. Examining the statistical difference between the two conditions,

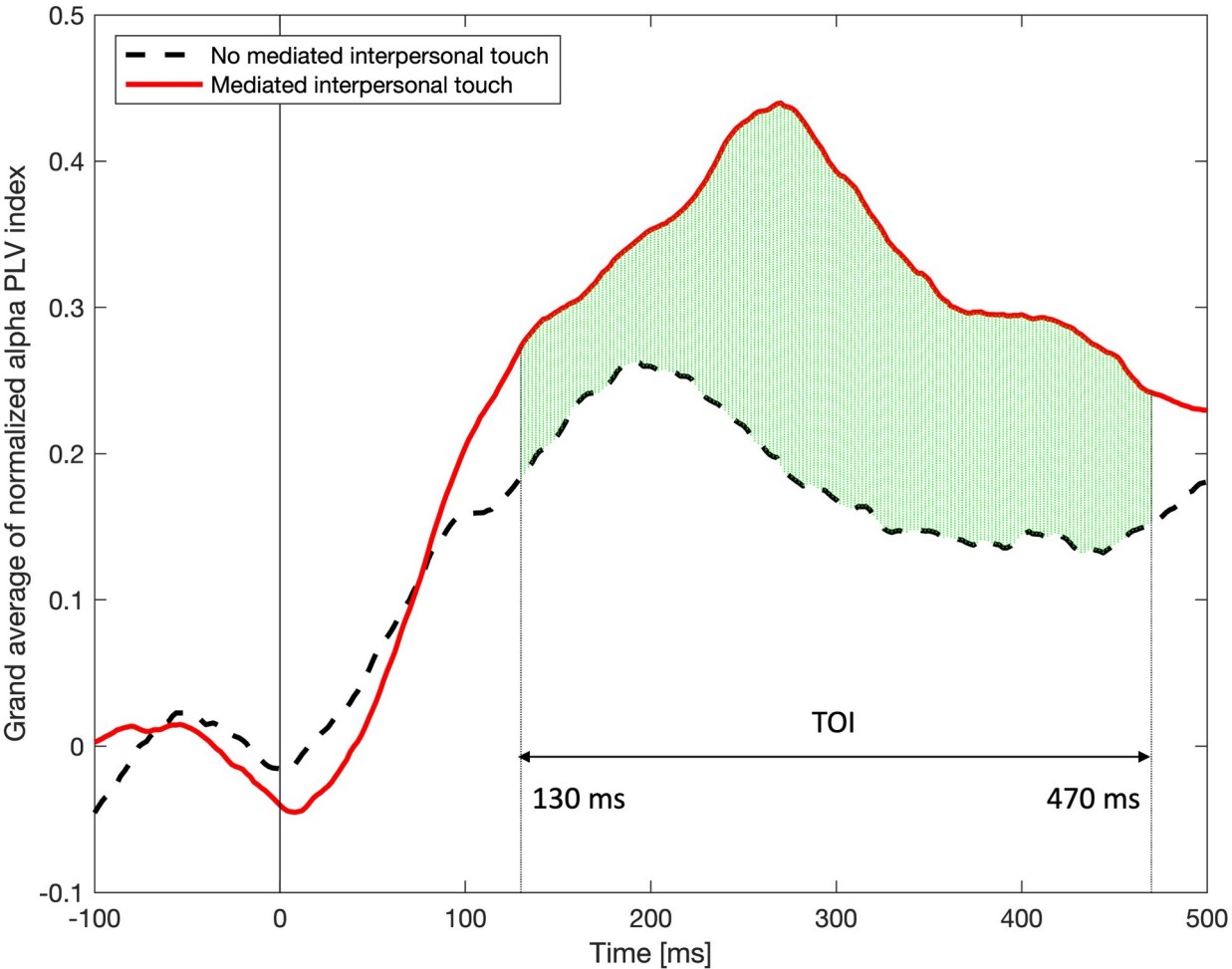

**Fig 6. Time of interest (130 ms to 470 ms) showing differences in normalized alpha PLV index from the whole brain area between no MIT and MIT interaction conditions.** Green vertical lines represent differences in PLV indices between no MIT and MIT conditions (Wilcoxon signed-rank test, corrected $p < 0.01$ by the false discovery rate). No significant differences are observed beyond 500 ms.

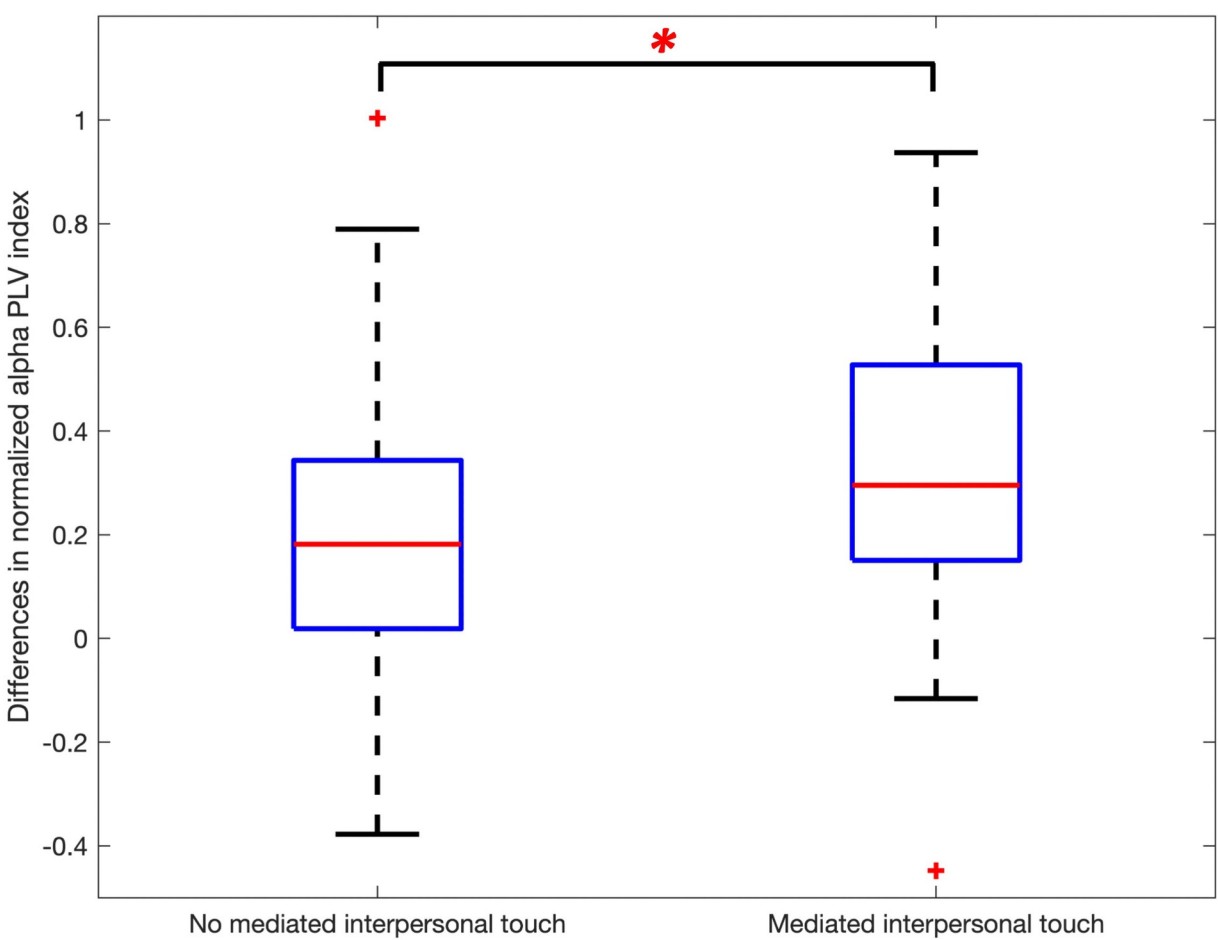

**Fig 7. Differences in alpha PLV index between no MIT and MIT conditions (paired t-test, $t(28)$ = 2.3148, \*$p$ = 0.0282).**

significant differences are found for the time interval from 130 ms to 470 ms (Wilcoxon signed-rank test, corrected $p < 0.01$ by the false discovery rate). Thus, we found this time interval (130–470 ms) and call the TOI. Fig 7 shows a significant difference when statistically comparing the changes in whole-interbrain functional connectivity between no MIT and MIT conditions. The normalized alpha PLVs of MIT task are significantly higher than the normalized alpha PLVs of no MIT task (paired t-test, $t(28)$ = 2.3148, $p$ = 0.0282).

We also investigated how the interbrain functional connectivity changes compared to the baseline (1 second before cue) within the TOI in the no MIT and the MIT conditions. The average PLV index in the TOI was compared to the baseline for the EEG channels of the two participants. When the average of the PLV indices between the EEG channels of two participants during TOI increased compared to the baseline, it was expressed in red solid lines, and when it decreased, it was expressed in a blue dashed line in Fig 8 of S1 Appendix. It can be seen that the interbrain PLVs increased for many channel pairs in the MIT task compared to the baseline, but not in the no MIT task.

## Discussion

In this study, we developed a system that enables mediated touch interaction through a cuff squeeze task between two people using a common air channel. The squeeze task was carefully

designed to provide a bidirectional flow of haptic information since the MIT encoder is also able to feel the air pressure at the same time as they squeeze the cuff. This is similar to direct interpersonal touch where the person applying the touch can feel the touch at the same time as the other person receiving the touch, thus creating a more realistic interpersonal touch experience. We hypothesized that interbrain synchrony would increase during mediated touch interaction between two participants and investigated whether there was a difference between no MIT and MIT conditions. We got a very interesting result: a significant increase in alpha interbrain synchronization between participants' brains during the early stage (130–470 ms) of the MIT interaction (Wilcoxon signed-rank test, corrected $p < 0.01$ by the false discovery rate). We presume that the combination of hyperscanning methods and haptic technologies could make dramatic contributions to our understanding of mediated or direct interpersonal touch. This study calls for hyperscanning as a more ecologically valid way of studying the neural correlates associated with MIT as it involves studying the interaction between multiple brains. It is clear that we need more interactive paradigms (to simulate more realistic and natural interpersonal touch interactions) and neuroimaging data coming from multiple brains to fully understand the brain's interpersonal touch system, both direct and mediated.

It is very interesting finding that the TOI is found to be from 130 ms to 470 ms (Fig 6). Furthermore, the peak of alpha PLV during MIT interaction occurs at around 270 ms, which happens even before the squeeze pressure is perceived by the MIT decoder (Fig 4). Also, in the topographies of Fig 5, no decrease in the alpha power of the motor and somatosensory cortex of the decoder was observed before 400 ms, and a similar pattern between the encoder and decoder was not observed before 400 ms. Nevertheless, in the case of MIT interaction, the fact that the PLV index between the MIT encoder and the MIT decoder increases at a very early stage implies that the two brains reacted in a coordinated fashion. In fact, this is the time the encoder prepares or initiates motor execution and cognitive functions, such as attention to the squeeze task. Given the role that alpha interbrain synchrony plays during social interaction [36], a significant increase in PLV index during MIT interaction goes beyond attention and seems to indicate an effect of social coordination.

Synchronized behavior, in the form of the squeeze profile, was observed between participants during MIT interaction. This is reflected in the decrease in the differences between the squeeze onset, squeeze energy, and squeeze speed parameters of the squeeze task from the start of the interaction where no social coordination is expected until the end of the session where social coordination is developed. MIT interaction seems to indicate a transition from independent to coordinated squeeze behavior. It is presumed that the differences in the squeeze behavior of the two participants at the end of the experimental session may have decreased due to a potential increase in social cooperation between the two communicating parties [44]. Even though no significant differences in the behavioral similarities from the first four to the last four trials are observed, similar findings are reported in the literature based on visual [45] and auditory [46] interaction.

Previous literature on EEG hyperscanning during social interaction [36] showed interbrain synchronization in the alpha, beta, and gamma bands. In this study, we also examined these frequency bands and found significant results only in the alpha band. This may be due to the different tasks. In mediated touch interaction through the hand, synchronization of the alpha PLV was the dominant feature because the alpha band is an important feature associated with social interaction [47]. Alpha and beta frequency bands are highly correlated with motor movements and we also expected results in the beta frequency band. However, the existing literature shows that alpha and beta functional connectivity between brains depend more on the interaction between two people than on the motor movement itself [35].

Although the findings of the present study demonstrate how MIT can elicit interbrain synchrony, a few limitations should be mentioned. The squeeze interface does not provide an accurate simulation of interpersonal touch. For example, the lack of thermal exchange, skin-to-skin contact, and visual/auditory interaction may decrease the fidelity of interpersonal touch and thus the overall user experience. Furthermore, MIT is strongly dependent on contextual factors that make generalization difficult [48]. Therefore, it is desirable to study interbrain synchrony under various interpersonal contexts (relationships, gender, role, testing environment, etc.). Finally, even though the MIT encoder receives haptic feedback while sending the squeeze, a task that involves the simultaneous, active exchange of interpersonal touch is desirable to develop a deeper understanding of interbrain synchrony during MIT. Gender effects were not analyzed in this study. Although 30 men and 30 women participated in the study, the sample size was insufficiently balanced to allow for gender effects analysis.

In summary, we utilized an EEG-based hyperscanning technique using the PLV index to analyze interbrain synchrony during MIT. Our results suggest that MIT elicits a significant increase in the alpha PLV index between participants' brains during the early stage of the task period. Furthermore, the squeeze behavior demonstrated increased similarity between the communicating parties and thus indicated a tendency for social coordination.

For future work, developing more realistic MIT interfaces, and possibly including other sensory modalities such as sound, vision, or even smell, would significantly improve the quality of interpersonal touch and eventually contribute to a better understanding of the neural correlates associated with MIT. In this study, we tried to minimize the amount of familiarity between the two participants in the experiment. This is because there is a study [49] that shows that the familiarity between the communicating parties has a clear impact. More research is needed on the differences between familiar and unfamiliar participants in mediated hand interaction. Furthermore, we plan to investigate the effects of MIT in different realistic contexts, such as different types of interpersonal touch (hug, handshake, shoulder tap, etc.), different relationships, or gender effects. Finally, studying the effects of MIT in a longitudinal explorative field study will shed light on the long-term benefits/challenges for end-users.

## Supporting information

**S1 Appendix. Alpha interbrain synchrony.**
(PDF)

## Author Contributions

**Conceptualization:** Wanjoo Park, Mohamad Eid.

**Data curation:** Wanjoo Park, Muhammad Hassan Jamil.

**Formal analysis:** Wanjoo Park.

**Funding acquisition:** Mohamad Eid.

**Investigation:** Wanjoo Park, Mohamad Eid.

**Project administration:** Wanjoo Park.

**Software:** Muhammad Hassan Jamil.

**Supervision:** Mohamad Eid.

**Validation:** Wanjoo Park.

**Writing – original draft:** Wanjoo Park, Muhammad Hassan Jamil, Mohamad Eid.

**Writing – review & editing:** Wanjoo Park, Muhammad Hassan Jamil, Mohamad Eid.

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
