## [Decision Letter · Decision Letter 0]

17 Oct 2023

PONE-D-23-09147Alpha Interbrain Synchrony During Mediated Interpersonal TouchPLOS ONE

Dear Dr. Eid,

Thank you for submitting your manuscript to PLOS ONE. After careful consideration, we feel that it has merit but does not fully meet PLOS ONE’s publication criteria as it currently stands. Therefore, we invite you to submit a revised version of the manuscript that addresses the points raised during the review process.

We look forward to receiving your revised manuscript.

Kind regards,

Sandra Carvalho, Ph.D.

Academic Editor

PLOS ONE

Journal Requirements:

a) Did participants provide their written or verbal informed consent to participate in this study?

b) If consent was verbal, please explain i) why written consent was not obtained, ii) how you documented participant consent, and iii) whether the ethics committees/IRB approved this consent procedure

"This work is supported by the NYUAD Center for Artificial Intelligence and Robotics(CAIR), funded by Tamkeen under the NYUAD Research Institute Award,"

"YES. M.E. received the funding. This work is supported by the NYUAD Center for Artificial Intelligence and Robotics (CAIR), funded by Tamkeen under the NYUAD Research Institute Award, CG010. URL for funding agency: https://www.tamkeenuae.com/"

Reviewers' comments:

Reviewer's Responses to Questions

**Comments to the Author**

1. Is the manuscript technically sound, and do the data support the conclusions?

Reviewer #1: Partly

Reviewer #2: No

2. Has the statistical analysis been performed appropriately and rigorously? 

Reviewer #1: Yes

Reviewer #2: No

3. Have the authors made all data underlying the findings in their manuscript fully available?

Reviewer #1: Yes

Reviewer #2: No

4. Is the manuscript presented in an intelligible fashion and written in standard English?

Reviewer #1: Yes

Reviewer #2: Yes

5. Review Comments to the Author

Reviewer #1: In this study, the researchers examined the neural synchrony during mediated touch. Paris of strangers each pressed and received pressure from a sleeve and their neural activity was measured. Researchers found increased synchrony in the alpha band during squeezes. This study is well designed and well analyzed. I think the results can contribute to the field of mediated touch and its neural underpinnings. However, as the manuscript is written now it is unfit for publication. The authors need to elaborate and pick better references for their claims in the introduction section. They need to better define their hypothesis and add about their task and why they chose to design it. The whole discussion section is written as a technical description of the task and the results with almost no mention of the meaning of the results, their significance, and their contribution to the field. It is almost written as a review of a new technological device and not an academic paper.

Abstract:

1. Please add a sentence or two about interpreting your findings and their significance.

2. Add abbreviation (PLV) after introducing the term for the first time.

Introduction:

1. There is a mismatch between some of the papers the authors cited to the claim they make in the introduction:

a. Citing the Eid et al., 2015 when talking about intimacy of touch is not right here since this paper is about haptic touch – it is more appropriate later when talking about haptic touch.

b. Citing the Dunbar 1991 when talking about handshaking in social interactions is not right since this paper is about back grooming in primates and how it is social and not just hygienic – there are more suitable papers on human touch to cite here.

c. Erl et al., 2015 talks about mediated social touch and does not talk about “kissing hugging handholding or cuddling” in any way.

2. “touch can convey vitality and immediacy more powerfully than language” – can the authors elaborate on that? Especially since it may be connected to their topic of the study.

3. “touch is commonly utilized to enhance the meaning of other forms of verbal and non-verbal communications” – please elaborate on that as well here.

4. Please explain the Midas touch effect for the readers who are not familiar with the literature on that

5. Please explain the “negative connotations” of touch you mentioned.

6. Please cite Goldstein et al., 2018 along with your citation of Kaus et al., 2020 – about touch and hyperscanning.

Goldstein, P., Weissman-Fogel, I., Dumas, G., & Shamay-Tsoory, S. G. (2018). Brain-to-brain coupling during handholding is associated with pain reduction. Proceedings of the national academy of sciences, 115(11), E2528-E2537.

7. Explain what is “phase locking value” and why is it good to use it here to measure your variables.

8. Be more specific about your hypothesis – why would MIT increase interbrain synchrony? What are the measures that will be larger? Compared to what will it be larger? Please justify your study and its meaning and significance more in the introduction. At the moment it is a bit lacking although I do think this topic is of interests, its just needs to be expressed better.

Methods:

1. Please provide mean and s.d of age of the participants.

2. What is the rationale for having participants who are unfamiliar with one another?

3. Were the pairs mixed gender or same gender? Or both?

4. State that the sample is composed of 30 pairs

5. Regarding the behavioral analysis – what is the rationale behind looking at the timing of the squeezes and how they change over time? The authors communicated to the reader that the objective of this study is to look at neural bases of mediated touch. Please explain and, if appropriate, add a section in the introduction about the meaning of this.

Results:

6. “It is presumed that prefrontal alpha oscillation increases due to recognizing the 223

role of the MIT encoder and preparing for the motor task” – please provide a reference for this assumption.

Discussion:

“First of all, it is worth noting that the MIT is well perceived by the communicating 286

participants. When asked whether they perceived the squeeze or not, the MIT decoder 287

confirmed that they felt the squeeze cue more than 95 % of the times (mean 98.28 %, 288

SD 2.91 %). Furthermore, during the no MIT task, both participants confirmed that no 289

squeeze is felt more than 95 % of the time (mean 95.95 %, SD 6.74 %)). This clearly 290

demonstrates that the squeeze MIT was reliably communicated and perceived between 291

the two participants”.

Please move this paragraph to the results section, discussion should be about interpreting results and not reporting them for the first time there.

“

differences in the squeeze behavior of the two participants at the end of the 313

experimental session may have decreased due to the increase in social coordination 314

between the two communicating parties

“

This is a very interesting finding – can the authors please elaborate why would social coordination decrease the difference in the squeeze? Please cite relevant studies to support this hypothesis.

The discussion is very descriptive of the results and does not interpret them nor compares the advancement made here for past data.

This whole section should be written differently with stating the purpose of the study and the results, then discuss past studies that relate to the present study and say what this study adds to them (studies about neural synchronization of touch and neural synchronization in general), then discuss why the alpha band had the most significant results (simulation mechanisms, empathy, social cognition, social skills – it is all in the literature and there are many studies on this topic), then mention limitations and the study and then have the last paragraph.

Discuss the role of gender since you recruited both men and women for the study

Discuss the topic of familiarity since there is literature on synchrony between familiar and stranger pairs.

Reviewer #2: Wanjoo Park et al. analyzed brain synchronization between 2 participants using EEG data while one of them had to squeeze a cuff and the other one received that pressure.

The authors found differences in alpha PSD and connectivity using the PLV index. Overall the manuscript is well written, but more information on methods is needed. The lack of correction for multiple comparisons and other methodological problems make it unsuitable for publication at this time.

My comments are included below.

Abstract: "The findings and limitations of this study are further discussed, and perspectives on future research are provided." Instead of this statement, the main conclusion of the work should be included in the abstract.

Hypothesis: No specific hypotheses are stated other than "We hypothesize that MIT increases interbrain synchrony" Therefore, it is supposed to be an exploratory work that should be accompanied by multiple comparisons correction.

Methods: Methods are written in the present tense. They are usually written in the past tense.

Regarding the experimental setup, it would be interesting to indicate what is the delay from the time the pressure sensor detects changes in the sender's cuff until the other participant receives changes in pressure. In addittion, could you please indicate the pressure resolution in the decoder participant?

From the picture it does not look like the participants were wearing headphones to isolate them from external noise. Did the setup (especially the air pump or valves) produce a lot of noise that could affect the participants, and in turn synchronize their brain activity with such noise?

Fig 3: please indicate the duration of the auditory cues.

Line 100: "The application is also responsible to record and store the pressure profile of the participants during the experimental task in a data file". I understand that you have the pressure profiles of both the "encoder" and the "decoder" participant, could you show the profiles in figure 4 separating decoder and encoder? this would help to know the delay between both.

Line 182: "Topography analysis for the four frequency bands was conducted". It is not clear at which scalp locations the PSDs of the different frequency bands are compared. The authors might consider performing a time-frequency analysis instead of PSDs every 200 ms. In general, it would be necessary to go deeper into how the analyses were performed.

Lines 190-193: This sounds like double-dipping. The authors first look for the time window with statistical differences and then analyze this time window a second time. I believe that this is not a correct way to perform the analysis.

Results: Again, the absence of correction for multiple comparisons is perhaps the most limiting factor of the manuscript. Many comparisons are performed without specific hypotheses and there is no correction for false positives.

Fig 6: This figure should include the whole interval, like the other figures, up to 2500ms. How is it possible that there are differences between MIT and non-MIT before somatosensory stimulation occurs? In the discussion the authors interpret this as an index of social coordination. However, it looks to me like an index of readiness for a condition in which they must perform a task. Probably the brain activity of both participants are synchronized to the auditory cue in the active condition, which requires more attentional resources than in the non-MIT condition, in which participants do not have to do anything. Makes more senso to me that such synchronization is the result of both participants reacting to the sound of the cue, irrespective of social interaction.

In the interpretation of Fig 7: "When the average of the PLV indices between the EEG channels of two participants during TOI increased compared to the baseline, it was expressed in red solid line, and when it decreased, it was expressed in a blue dashed line in Figure 7. It can be seen that the interbrain functional connectivity increased significantly in the case of MIT compared to the case of No MIT" The authors should be careful with this interpretation, they have not compared between conditions, only TOI vs baseline, thefore they cannot say that there were significant differences between conditions.

Line 266: t and p-value should be included.

Discussion: Alpha and beta are two key somatosensory frecuencies. The authors could discuss at some point why no PSD differences are observed in beta.

Line 287: It would be more appropriate to include this information in the results than in the discussion.

6. PLOS authors have the option to publish the peer review history of their article (what does this mean?). If published, this will include your full peer review and any attached files.

Reviewer #1: No

Reviewer #2: No

---

## [Author Response · Author response to Decision Letter 0]

22 Nov 2023

Please see the Response To Reviewers document included in the submission.

---

## [Decision Letter · Decision Letter 1]

22 Feb 2024

Alpha Interbrain Synchrony During Mediated Interpersonal Touch

PONE-D-23-09147R1

Dear Dr. Eid,

We’re pleased to inform you that your manuscript has been judged scientifically suitable for publication and will be formally accepted for publication once it meets all outstanding technical requirements.

Kind regards,

Ali Mohammad Alqudah

Academic Editor

PLOS ONE

Additional Editor Comments (optional):

Reviewers' comments:

Reviewer's Responses to Questions

**Comments to the Author**

1. If the authors have adequately addressed your comments raised in a previous round of review and you feel that this manuscript is now acceptable for publication, you may indicate that here to bypass the “Comments to the Author” section, enter your conflict of interest statement in the “Confidential to Editor” section, and submit your "Accept" recommendation.

Reviewer #3: All comments have been addressed

Reviewer #4: All comments have been addressed

2. Is the manuscript technically sound, and do the data support the conclusions?

Reviewer #3: Yes

Reviewer #4: Yes

3. Has the statistical analysis been performed appropriately and rigorously? 

Reviewer #3: Yes

Reviewer #4: Yes

4. Have the authors made all data underlying the findings in their manuscript fully available?

Reviewer #3: Yes

Reviewer #4: Yes

5. Is the manuscript presented in an intelligible fashion and written in standard English?

Reviewer #3: Yes

Reviewer #4: Yes

6. Review Comments to the Author

Reviewer #3: I am satisfied with the changes incorporated in the revised manuscript and recommend it for publication.

Reviewer #4: Dear Authors,

I would like to thank you for this very good and interesting paper. The paper is one of the most interesting papers that I have read since I started reviewing for Plos One. The authors have addressed al comments.

7. PLOS authors have the option to publish the peer review history of their article (what does this mean?). If published, this will include your full peer review and any attached files.

Reviewer #3: **Yes: **Braj Bhushan

Reviewer #4: **Yes: **Mai Abdel Haleem A. Abusalah

---

## [Editor Report · Acceptance letter]

25 Mar 2024

PONE-D-23-09147R1 

PLOS ONE

Dear Dr. Eid, 

I'm pleased to inform you that your manuscript has been deemed suitable for publication in PLOS ONE. Congratulations! Your manuscript is now being handed over to our production team.

Kind regards, 

on behalf of

Dr. Ali Mohammad Alqudah 

Academic Editor

PLOS ONE